# Selective Localization of Carbon Black in Bio-Based Poly (Lactic Acid)/Recycled High-Density Polyethylene Co-Continuous Blends to Design Electrical Conductive Composites with a Low Percolation Threshold

**DOI:** 10.3390/polym11101583

**Published:** 2019-09-27

**Authors:** Xiang Lu, Benhao Kang, Shengyu Shi

**Affiliations:** Key Laboratory of Polymer Processing Engineering of the Ministry of Education, National Engineering Research Center of Novel Equipment for Polymer Processing, Guangdong Key Laboratory of Technique and Equipment for Macromolecular Advanced Manufacturing, South China University of Technology, Guangzhou 510641, China; luxiang_1028@163.com (X.L.); hnlgkbh@163.com (B.K.)

**Keywords:** poly (lactic acid), high-density polyethylene, carbon black, co-continuous, selective localization

## Abstract

The electrically conductive poly (lactic acid) (PLA)/recycled high-density polyethylene (HDPE)/carbon black (CB) composites with a fine co-continuous micro structure and selective localization of CB in the HDPE component were fabricated by one-step melt processing via a twin-screw extruder. Micromorphology analysis, electrical conductivity, thermal properties, thermal stability, and mechanical properties were investigated. Scanning electron microscope (SEM) images indicate that a co-continuous morphology is formed, and CB is selectively distributed in the HDPE component. With the introduction of CB, the phase size of the PLA component and the HDPE component in PLA/HDPE blends is reduced. In addition, differential scanning calorimetry (DSC) and thermos gravimetric analysis (TGA) results show that the introduction of CB promotes the crystallization behavior of the PLA and HDPE components, respectively, and improves the thermal stability of PLA70/30HDPE/CB composites. The electrically conductive percolation threshold of the PLA70/30HDPE/CB composites is around 5.0 wt %, and the electrical conductivity of PLA70/30HDPE/CB composites reaches 1.0 s/cm and 15 s/cm just at the 10 wt % and 15 wt % CB loading, respectively. Further, the tensile and impact tests show that the PLA70/30HDPE/CB composites have good mechanical properties. The excellent electrical conductivity and good mechanical properties offer the potential to broaden the application of PLA/HDPE/CB composites.

## 1. Introduction

Compared with synthesizing a new polymer, the blending of various thermoplastic polymers is an important and efficient technique to developing high performance polymeric materials [1,2]. Owing to thermodynamic reasons, most polymer blends are immiscible and tend to separate into two or more distinct phases during the processing. For two-phase blends, “sea-island” and co-continuous micro structure are the two major phase morphologies [3,4,5]. The co-continuous morphology of two-phase blends consists of two-coexisting, continuous, and interconnected phases throughout the entire blend volume [6,7,8,9]. Polymer blends with a co-continuous structure have many interesting applications and excellent performance, such as electrical conductivity, thermal conductivity, heat resistance, and so on [5,9].

In recent decades, electrical conductive polymer composites (ECPCs) have aroused enormous attention in various high value-added applications, such as anti-static packaging materials, electromagnetic interference shielding materials, sensors, and conductors [8,10,11,12] ECPCs usually consist of a conductive filler and an insulating matrix. In order to achieve excellent electrical conductivity, it is necessary to incorporate enough conductive filler to form a continuous conductive network. The electrically conductive percolation theory is commonly used to describe the insulator-to-conductor transition in electrically conductive polymer composites. Further, the electrical conductive percolation threshold is considered to be the minimum electrical conductive filler content to form a continuous electrical conductive network [13]. Compared with the single-polymer ECPCs with a high electrical conductive filler content, designing two-phase immiscible polymer blends with co-continuous micro structure and selective localization of the electrical conductive filler in only one phase or at the phase interface is an effective approach to reduce the electrical conductive percolation threshold [14,15,16,17,18,19,20]. For instance, Goldel et al. [19] introduced multiwalled carbon nanotubes (MWCNTs) into the immiscible and co-continuous polycarbonate (PC) and poly (styrene-acrylonitrile) (SAN) blends with the MWCNTs selectively located within the PC component, the special microstructure resulted in much lower electrical resistivities for PC/SAN/MWCNTs composites. Especially, selective localization of the electrically conductive filler at the blend interface is considered to be the more ideal way to decrease the percolation threshold. Gubbels et al. [14] prepared the carbon black (CB)/polystyrene (PS)/polyethylene (PE) ECPCs with a low percolation threshold by controlling the migration of CB to the interface of the PS/PE blend during the blending. Huang et al. [20] prepared MWCNTs/poly (lactic acid) (PLA)/poly (caprolactone) (PCL) ECPCs with an ultralow percolation threshold by controlling the migration process of MWCNTs at a continuous interface of the PLA/PCL blend. 

With increasing attention to the environmental protection and sustainable development, the biorenewable and biodegradable PLA has attracted significant interest from ecological perspectives in recent years [21,22,23]. At the same time, high-density polyethylene (HDPE) is widely used in flexible packaging and containers owing to its excellent performance and cheap price [24,25]. Also, the widespread use of traditional petroleum-based HDPE has produced a large amount of plastic waste and caused serious environmental problems. How to recycle and use this wasted HDPE has also attracted great attention. 

In our previous research [26], we discussed the phase morphology of PLA/HDPE blends with the increasing HDPE content. When the content of HDPE is between 30 wt % and 50 wt %, the PLA/HDPE blends with a stable co-continuous structure were obtained. This is a green and environmentally friendly approach to obtain electrically conductive PLA/HDPE/CB composites with the co-continuous micro structure and CB selective localization by blending biodegradable PLA, recycled HDPE, and CB. Moreover, the recycling and reuse of HDPE can also greatly reduce the use of new petroleum-based polymers. In this study, we prepared PLA/HDPE blends with a co-continuous structure and studied the effect of CB on the phase structure, electrical conductivity, thermal properties, and mechanical properties of these blends.

## 2. Experimental Method

### 2.1. Materials

PLA (4032D) was obtained from Natureworks, LLC (Minnetonka, MN, US). The recycled high-density polyethylene (MFI: 10 g/10 min, 190 °C, 2.16 kg) without any filler was obtained from the Kingfa Sci. & Tech. Co. Ltd., Guangzhou, China. A special electrically conductive grade of carbon black (CB) ENSACO 250 G from Timcal (Willebroek, Belgium), which was suitable for incorporation in thermoplastic polyolefin material, was used as electrically conductive filler. 

### 2.2. Preparation of PLA/HDPE/CB Electrically Conductive Composites

Firstly, the PLA and recycled HDPE pellets were dried in vacuum at 80 °C for 4 h to remove the absorbed moisture. A series of PLA/HDPE/CB electrically conductive composites was prepared via a twin-screw extruder at about 200 °C and 60 rpm. The screw diameter and length/diameter ratio were 25 mm and 20:1, respectively. Then, the extruded pellets were dried at 80 °C for more than 4 h, and were injection-molded into american society for testing materials (ASTM)-standard specimens at 200 °C and 55 MPa.

### 2.3. Characterization

The micro morphology of PLA/HDPE blends and PLA/HDPE/CB composites was imaged by a scanning electronic microscope (SEM, HITACHI SE3400N, Tokyo, Japan) at 10–15 kV accelerating voltage. In order to clearly distinguish the micro phase structure, the PLA phase was etched by chloroform for 2 h at room temperature. 

The static contact angle of compression-molded PLA, HDPE, and CB films was performed with an OCA 15 PLUS apparatus (Dataphysics Co. Ltd., Filderstadt, Germany), static contact angles of distilled water (H_2_O) and diiodomethane (CH_2_I_2_) were measured by depositing a drop of 3–5 mL on the sample surface, and the values were estimated as the tangent normal to the drop at the intersection between the sessile drop and the surface. All contact angles of a given sample were carried out at least five times.

The volume electrical conductivity of the PLA/HDPE/CB composites was measured by the four probe method. Silver paste was attached to test side of each sample to ensure good contacts between the samples and the electrodes. The dimension of the tested samples was 10 × 10 × 1.0 mm^3^.

The melting and cooling behaviors of PLA/HDPE/CB composites were investigated by differential scanning calorimetry (DSC, Netzsch 204c, Selb, Germany) equipped with a liquid nitrogen-cooling accessory between 30 °C and 200 °C at 10 °C /min under a nitrogen atmosphere.

The thermal stability of PLA/HDPE/CB composites was performed by thermogravimetric analysis (TGA, Netzsch TG209, Selb, Germany) between 30 °C and 700 °C under a 250 mL/min nitrogen and 10°C /min heating ramp.

The tensile strength, elongation at break, and impact strength of PLA/HDPE/CB composites were tested by an Instron universal machine (model 5566, Norfolk, MA, USA) and Instron POE2000 pendulum impact tester in accordance with International Organization for Standardization (ISO 527 and ISO 179^−1^), respectively. Five repeated tests were used to obtain the average of the tensile strength, elongation at break and impact strength.

## 3. Results and Discussion

### 3.1. Morphology of PLA/HDPE Blends

For polymer blends, the microscopic phase morphology plays a decisive role in the macroscopic properties [27]. Prior to the study of ternary PLA/HDPE/CB blends, it is necessary to investigate phase morphology development of the PLA/HDPE binary blends. Figure 1a–d shows the SEM images of fracture surface for the PLA/HDPE binary blends with different component ratios, respectively. For 50/50, 60/40, and 70/30 (PLA/HDPE, *w/w*) blends, a co-continuous phase morphology is observed. With the PLA content further increase to 80 wt %, a typical island-sea type morphology is observed, where discrete droplets of the minor phase (HDPE) are dispersed in the matrix (PLA). It is well known that the chloroform is a good solvent for PLA, but an inert solvent for HDPE. In order to better observe the co-continuous microscopic phase morphology of the PLA/HDPE binary blends (50/50, 60/40, 70/30), the PLA component of the PLA/HDPE binary blends was etched by chloroform, and the SEM images are shown in Figure 1a’–c’, respectively. From the SEM images of the etched samples, it can be seen that all the 50/50, 60/40, and 70/30 blends observed a typical co-continuous phase morphology. As we described in Section 1, the co-continuous structure of polymer blends has many interesting applications, such as electrical conductivity, thermal conductivity, and so on. In the following discussion, in order to use more environmentally friendly biodegradable materials (such as PLA) and to build a good electrically conductive network in co-continuous PLA/HDPE blends with a low percolation threshold, the blending ratio of the PLA/HDPE blend was fixed at 70/30.

### 3.2. Effect of CB on the Morphology of PLA/HDPE Blends

During processing, compared with binary blends, the phase morphology development of ternary blends is more complicated [28,29]. Especially for the electrical properties of composites with double percolation structure, the phase morphology of the co-continuous structure plays an important role. With the introduction of CB into the PLA/HDPE (70/30) blend, compared with the pure PLA/HDPE (70/30) blend, the viscosity ratio, interfacial tension, and some other parameters are changed significantly, and the finally phase morphology of the PLA/HDPE/CB (70/30/x) blend is also obviously affected by the CB loadings. Figure 2a–g shows the fracture surface SEM images of the PLA/HDPE/CB ternary blends with different CB loadings (1.5 wt %, 3.0 wt %, 5.0 wt %, 7.5 wt %, 10.0 wt %, 12.5 wt %, and 15.0 wt %), respectively. Furthermore, Figure 2a’–g’ shows the corresponding fracture surface SEM images of the PLA/HDPE/CB ternary blends that etched the PLA phase. As can be seen from the SEM images, for the current system, the co-continuous structure of the PLA/HDPE/CB (70/30/x) blend is not destroyed by the introduction of CB. Instead, with the introduction of CB, the phase morphology of the co-continuous structure of the PLA/HDPE/CB (70/30/x) ternary blend becomes better. With the CB loading increases, and the phase size of continuous HDPE phase becomes smaller, which will contribute to the improvement of mechanical properties and electrical properties of the PLA/HDPE/CB (70/30/x) ternary blend.

### 3.3. Selective Distribution of CB Particles

Generally, the localization of CB in immiscible PLA/HDPE blends is mainly determined by the combined action of thermodynamic and kinetic factors, and can be predicted by the wetting parameter $\omega_{12}$ according to Young’s equation (Equation (1)) [14]:(1)$$\omega =\frac{\gamma_{CB-HDPE}-\gamma_{CB-PLA}}{\gamma_{HDPE-PLA}},$$
where $\gamma_{CB-HDPE}$ stands for interfacial tension between CB and HDPE, $\gamma_{CB-PLA}$ stands for interfacial tension between CB and PLA, and $\gamma_{HDPE-PLA}$ stands for interfacial tension between HDPE and PLA. Depending on the value of ω, CB tends to be localized in HDPE (ω < −1), in PLA (ω > 1), or at the interface between HDPE and PLA (−1 < ω < 1). According to the harmonic-mean equation (Equation (2)) and geometric-mean equation (Equation (3)) [30,31], the interfacial tension (γ) between different components can be calculated:(2)$$\gamma_{12}=\gamma_{1}+\gamma_{2}-4\left(\frac{\gamma_{1}^{d}\gamma_{2}^{d}}{\gamma_{1}^{d}+\gamma_{2}^{d}}+\frac{\gamma_{1}^{p}\gamma_{2}^{p}}{\gamma_{1}^{p}+\gamma_{2}^{p}}\right),$$
(3)$$\gamma_{12}=\gamma_{1}+\gamma_{2}-2\left(\sqrt{\gamma_{1}^{d}\gamma_{2}^{d}}+\sqrt{\gamma_{1}^{p}\gamma_{2}^{p}}\right),$$
where $\gamma_{1}^{d}$ and $\gamma_{1}^{p}$ stand for the dispersive and polar parts of the surface tension of component 1, respectively; and $\gamma_{2}^{d}$ and $\gamma_{2}^{p}$ stand for the dispersive and polar parts of the surface tension of component 2, respectively. $\gamma_{1}$ and $\gamma_{2}$ are the surface energy of component 1 and component 2, respectively. Contact angle measurement is a traditional method to calculate the surface energy of solids. According to Fowkes and his co-workers’ research results (Equation (4)) [32] and the Owens–Wendt equation (Equation (5)) [33],
(4)$$\gamma =\gamma^{d}+\gamma^{p},$$
(5)$$\gamma_{l}\left(1+\cos\theta \right)=2\left(\sqrt{\gamma_{s}^{d}\gamma_{l}^{d}}+\sqrt{\gamma_{s}^{p}\gamma_{l}^{p}}\right),$$
where $\gamma_{s}$ and $\gamma_{l}$ stand for the surface energy of the solid and liquid, respectively; θ stands for the contact angle; and $\gamma_{s}^{d}$, $\gamma_{s}^{p}$, $\gamma_{l}^{d}$, and $\gamma_{l}^{p}$ stand for the dispersive and polar components of the solid and liquid, respectively. According to the known $\gamma_{l}^{d}$ and $\gamma_{l}^{p}$ parameters of polar liquid and nonpolar liquid (H_2_O: $\gamma_{H_{2}O}^{p}$ = 50.8 MJ/m^2^ and $\gamma_{H_{2}O}^{d}$ = 22.5 MJ/m^2^; CH_2_I_2_: $\gamma_{CH_{2}I_{2}}^{p}$ = 2.3 MJ/m^2^, and $\gamma_{CH_{2}I_{2}}^{d}$ = 48.5 MJ/m^2^) [34] and the corresponding contact angle, $\gamma_{s}^{d}$, $\gamma_{s}^{p}$, and $\gamma_{s}$ can be calculated by combining Equations (4) and (5). The digital photos of water and the CH_2_I_2_ contact angle for the HDPE, PLA, and used CB, as well as the calculated surface parameters, are listed in Table 1.

On the basis of the calculated surface parameters of HDPE, PLA, and CB from Table 1, according to Equations (2) and (3), the $\gamma_{CB-HDPE}$, $\gamma_{CB-PLA}$, and $\gamma_{HDPE-PLA}$ are 0.8, 11.7, and 7.2 MJ/m^2^ (Equation (2)) and 0.4, 6.4, and 3.9 MJ/m^2^ (Equation (3)), respectively. According to Equation (1), ω is −1.5 (harmonic-mean equation) or −1.5 (geometric-mean equation), respectively. Therefore, the theoretically thermodynamic calculation indicates that CB tends to be selectively located in the HDPE phase during the melt blending process.

Figure 3 shows the SEM images of CB and PLA/HDPE/CB (70/30/10) composites. As shown in Figure 3a, the CB is a fluffy powder. In addition, as described above, the chloroform is a good solvent for PLA, but an inert solvent for HDPE. By comparing Figure 3b and Figure 3c, the smooth cryo-fracture surface in Figure 3b can be attributed to the PLA phase, and the ragged cryo-fracture surface in Figure 3b should be attributed to the HDPE phase. From Figure 3b,c, it can be seen that the fluffy CB powder is dispersed homogeneously in the HDPE phase without obvious aggregation, but no CB is observed in the PLA phase. The SEM results are consistent with the above analysis that CB tends to be selectively located in the HDPE component of PLA/HDPE blends. Considering the co-continuous structure of the PLA/HDPE/CB composites and the selective localization of CB in the HDPE component, the PLA/HDPE/CB composites should have excellent electrical conductivity.

### 3.4. Electrical Conductivity

In order to illustrate the excellent electrical conductivity of PLA/HDPE/CB composites owing to the co-continuous structure and selectively located CB, the CB-filled HDPE with different CB loadings was prepared via the same processing conditions as a comparison. Figure 4a shows the volume fraction of CB in the HDPE phase of PLA70/30HDPE composites and HDPE with the same CB loadings. According to the following formula (Equation (6)):(6)m=V·ρ
where m is the sample weight, V is the sample volume, and ρ is the sample density (the density for CB, PLA, and HDPE is 1.8, 1.24, and 0.9 g/cm^3^, respectively). As shown in Figure 4a, the calculated results show that the volume fraction of CB in HDPE is 0.8%, 1.6%, 2.7%, 4.1%, 5.5%, 7.0%, 8.5%, 11.7%, 15.0%, and 18.2% for 1.5wt %, 3.0wt %, 5.0wt %, 7.5wt %, 10.0wt %, 12.5wt %, 15.0wt %, 20.0wt %, 25.0wt %, and 30.0wt % CB loadings, respectively. However, for PLA70/30HDPE/CB composites, the volume fraction of CB in the HDPE phase of PLA70/30HDPE/CB composites is 2.6%, 5.1%, 8.5%, 12.5%, 16.4%, 20.1%, and 23.7% for 1.5 wt %, 3.0 wt %, 5.0 wt %, 7.5 wt %, 10.0 wt %, 12.5 wt %, and 15.0 wt % CB loadings, respectively. It can be seen that the volume fraction of CB in the HDPE phase of PLA70/30HDPE/CB composites is much larger than that in HDPE for the same CB loadings. This will facilitate the PLA70/30HDPE/CB ternary blends to achieve higher electrical conductivity at lower CB levels. Figure 4b shows the electrical conductivity of HDPE/CB composites and PLA70/30HDPE/CB composites with different CB loadings. From Figure 4b, a significant jump in electrical conductivity is observed for HDPE/CB composites when CB content is higher than 15.0 wt %, which indicates that the electrically conductive percolation threshold of the HDPE/CB composites is approximately 15.0 wt %. The electrical conductivity of HDPE/CB composites reaches 1.2 s/cm when the CB loading is 30 wt %. However, for PLA70/30HDPE/CB composites, the electrical conductivity jumps at 5.0 wt %, which indicates that the electrical conductive percolation threshold of the PLA70/30HDPE/CB composites is approximately 5.0 wt %, and the electrical conductivity of PLA70/30HDPE/CB composites reaches 1.0 s/cm and 15 s/cm just at the 10 wt % and 15 wt % CB loading, respectively. It can be seen that the electrically conductive percolation threshold of PLA70/30HDPE/CB composites is much lower than that of HDPE/CB composites. Interestingly, for the HDPE/CB composite with 25 wt % CB loading, the volume fraction of CB in the HDPE/CB (75/25) composite is about 15 vol %. Further, for the PLA70/30HDPE/CB composite with 10 wt % CB loading, the volume fraction of CB in HDPE component is 16.4 vol %, which is similar to that in the HDPE/CB (75/25) composite. It is important is that the electrical conductivity of HDPE/CB (75/25) and PLA/HDPE/CB (70/30/10) is also very close. This indicates that the co-continuous micro structure and CB selective localization is a good approach to prepare the electrical conductive polymer composites with a low electrically conductive percolation threshold.

### 3.5. Thermal Properties

On the basis of the above results, as expected, the co-continuous structure of PLA70/30HDPE/CB composites and the selective distribution of CB in PLA70/30HDPE/CB composites have a significant impact on the electrical conductivity of the PLA70/30HDPE/CB composites. However, how does this particular microstructure affect the thermal performance of the PLA70/30HDPE/CB composite? The differential scanning calorimeter (DSC) and thermogravimetric analysis (TGA) were performed to investigate the influence of CB loading on the melting and crystallization behaviors and thermal stability of PLA70/30HDPE/CB composites. Figure 5a,b show the melting and cooling curves of PLA70/30HDPE/CB with different CB loadings, respectively. The corresponding thermal parameters, such as glass transition temperature of the PLA component ($T_{g-PLA}$), melting temperature of the PLA component ($T_{m-PLA}$), melting temperature of the HDPE component ($T_{m-HDPE}$)), melting enthalpy of the PLA component ($∆H_{m-PLA}$), melting enthalpy of the HDPE component ($∆H_{m-HDPE}$), crystallization temperature of the HDPE component ($T_{c-HDPE}$), relative crystallinity of the PLA component ($X_{C-PLA}$), and relative crystallinity of the HDPE component ($X_{C-HDPE}$), are listed in Table 2. $X_{C-PLA}$ and $X_{C-HDPE}$ are calculated as follows (Equations (7) and (8)).
(7)$$X_{c\left(PLA\right)}=\frac{∆H_{m-PLA}-∆H_{cc-PLA}}{\omega_{PLA}∆H_{m-PLA}^{o}}\times 100\%,$$
(8)$$X_{c\left(HDPE\right)}=\frac{∆H_{m-HDPE}}{\omega_{HDPE}∆H_{m-HDPE}^{o}}\times 100\%,$$
where $\omega_{PLA}$ and $\omega_{HDPE}$ are the mass fraction of PLA and HDPE in the PLA/HDPE/CB composites, respectively. Further, $∆H_{m-PLA}^{o}$ and $∆H_{m-HDPE}^{o}$ are the enthalpy of the original polymer crystal for PLA (93 J/g) [26] and HDPE (292 J/g) [25], respectively. $∆H_{cc-HDPE}^{o}$ is the cold crystallization enthalpy of the PLA component.

As shown in Table 2, the $T_{g-PLA}$, $T_{m-PLA}$, $T_{g-HDPE}$, and $T_{m-HDPE}$ do not change significantly with the introduction of CB. It is well known that the crystallization of the polymer from the molten state can be divided into two stages, homogeneous nucleation or heterogeneous nucleation and crystal growth. Owing to the selective dispersion of CB in the HDPE component of PLA70/30HDPE/CB composites and the fact that CB plays a role in promoting nucleation of the crystallization process of the HDPE component, theoretically, $T_{c-HDPE}$ should be significantly improved. However, there is no any significant change for $T_{c-HDPE}$. The reason may be that the regular HDPE segments have a good crystallization nucleation. As described above, with the introduction of CB, the phase size of the PLA component and the HDPE component in PLA/HDPE blends is reduced. Therefore, owing to the reduced phase size and the promoted crystallization behavior of the PLA component by the easily crystallized HDPE, $∆H_{CC-PLA}$ decreases and $X_{C-PLA}$ increases obviously, with the increasing CB loadings. At the same time, $X_{C-HDPE}$ also increases slightly with the increase of CB content owing to the crystal nucleation of CB.

The thermal stability is also improved for polymer composites. Figure 6a,b show the TGA and corresponding first derivative TGA (DTG) curves for PLA70/30HDPE/CB. The corresponding onset of degradation temperature ($T_{5}$, the temperature at 5 wt % loss), the maximum degradation temperature for the PLA component and the HDPE component (the peak temperature of the DTG curve), and the char formation at 600 °C are listed in Table 3. Figure 6 shows that the HDPE component has better thermal stability than the PLA component. After the introduction of CB into PLA70/30HDPE blend, the *T*_5_ of PLA70/30HDPE/CB composites increased with the increasing CB loading. It can be attributed to the reduced phase size between the PLA component and the HDPE component. At the same time, owing to the selective localization of CB in the HDPE component, $T_{max-PLA}$ increases from 465.3 °C to 471.7 °C and 478.1 °C with the 5 wt % and 10 wt % CB loadings, respectively. However, there is no significant change for $T_{max-PLA}$. These results indicate that the incorporation of HDPE and CB can improve the thermal stability of the PLA component in PLA/HDPE/CB composites.

It is very important for the functional composite to have a certain mechanical strength for a wide range of applications. The tensile strength, elongation at break, and impact strength of PLA70/30HDPE/CB composites with different CB loadings are shown in Figure 7 and Figure 8. As presented, the tensile strength of PLA70/30HDPE/xCB composites (x = 0, 1.5, 3, 5, 7.5, 10, 12.5) is increased from 38.6 MPa to 41.7 MPa with the increased CB loading. The similar tendency can be observed in other CB filled incompatible blends. It can be thought that the increase in the tensile strength is the result of the introduction of the higher modulus CB and the change of morphology between PLA and HDPE. In addition, compared with HDPE, PLA is a brittle polymer. Theoretically, the tensile toughness of PLA70/30HDPE blend is mainly determined by the PLA component and exhibits brittle tensile fracture with lower elongation at break. As shown in Figure 7, the elongation at break of the PLA70/30HDPE blend is just as low as 4.9%, which is very close to the elongation at break of pure PLA and much lower than that of pure HDPE. With the introduction of CB, and as CB is selectively distributed in the HDPE component, there is no change for the elongation at break of PLA70/30HDPE/CB composites with different CB loadings, and the elongation at break of PLA70/30HDPE/CB composites maintained at around 5% with the increasing content of CB. The reason is probably because CB is selectively dispersed in the tough HDPE phase, and the tensile toughness of the composite is still determined by the brittle PLA component. In the same way, with the increase of CB content, the impact strength of the composite is also maintained at about 5 kJ/m^2^, and there is no obvious change, as shown in Figure 8. Even so, the introduction of CB does not deteriorate the mechanical properties of the PLA/HDPE blend, but slightly increases the tensile strength of the PLA70/30HDPE/CB composite, and the mechanical properties of the PLA70/30HDPE/CB composite enable it to meet most application needs.

## 4. Conclusions

In summary, owing to the design of co-continuous structure and selective localization of CB, the electrically conductive PLA/HDPE/CB composites with a low percolation threshold and good mechanical properties are successfully prepared. With the introduction of CB into the PLA70/30HDPE blend, the co-continuous structure of PLA70/30HDPE/CB composites does not break with the increasing CB loadings. As expected, CB is selectively distributed in the HDPE component of PLA70/30HDPE/CB composites. It provides the possibility of achieving the low percolation threshold of the electrically conductive PLA/HDPE/CB composites. The electrical conductivity of PLA70/30HDPE/CB composites reaches 1.0 s/cm and 15 s/cm just at the 10wt % and 15 wt % CB loading, respectively, and the conductive percolation threshold of the PLA70/30HDPE/CB composites is just around 5.0 wt %, which is significantly lower than that of the HDPE/CB composites (15 wt %). In addition, the thermal stability and tensile strength of PLA70/30HDPE/CB composites are improved with the increasing CB loadings. All the results indicate that the obtained bio-based PLA70/30HDPE/CB electrically conductive composites with reliable and tunable properties have broad application prospects.

## Figures and Tables

**Figure 1 polymers-11-01583-f001:**
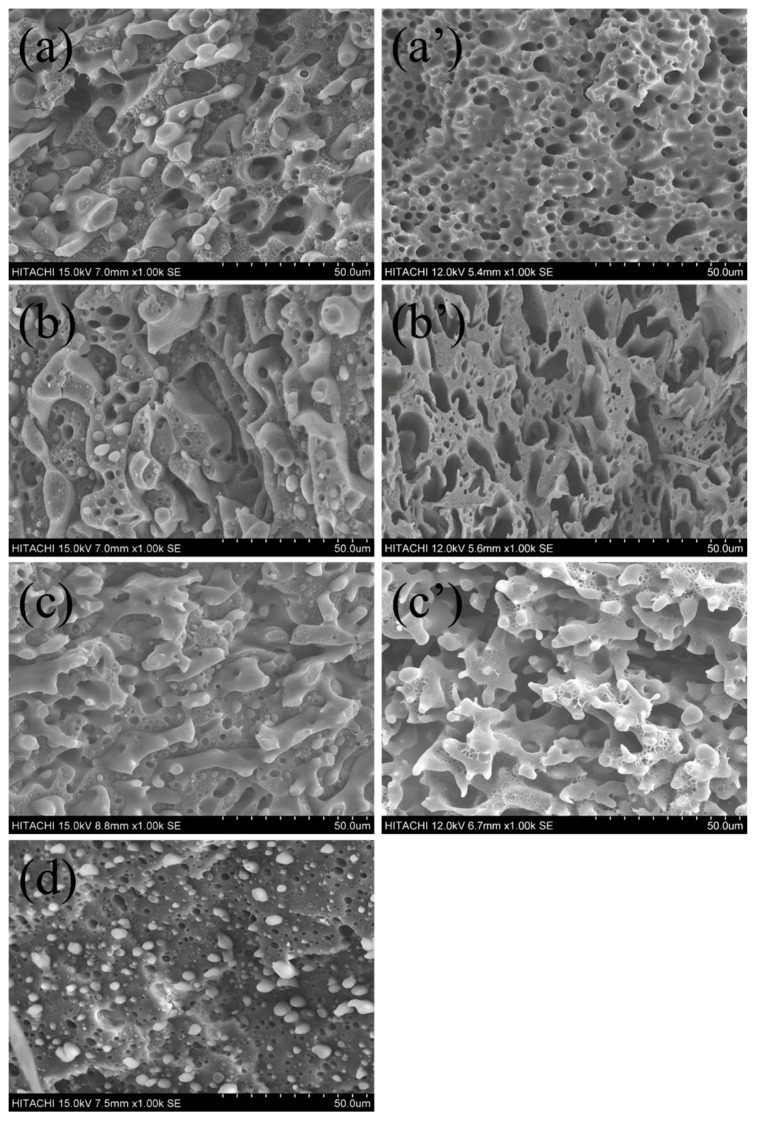
Scanning electron microscope (SEM) of poly (lactic acid) (PLA)/high-density polyethylene (HDPE) (*w/w*) blends, (**a**,**a′**) 50/50, (**b**,**b′**) 60/40, (**c**,**c′**) 70/30, and (**d**) 80/20.

**Figure 2 polymers-11-01583-f002:**
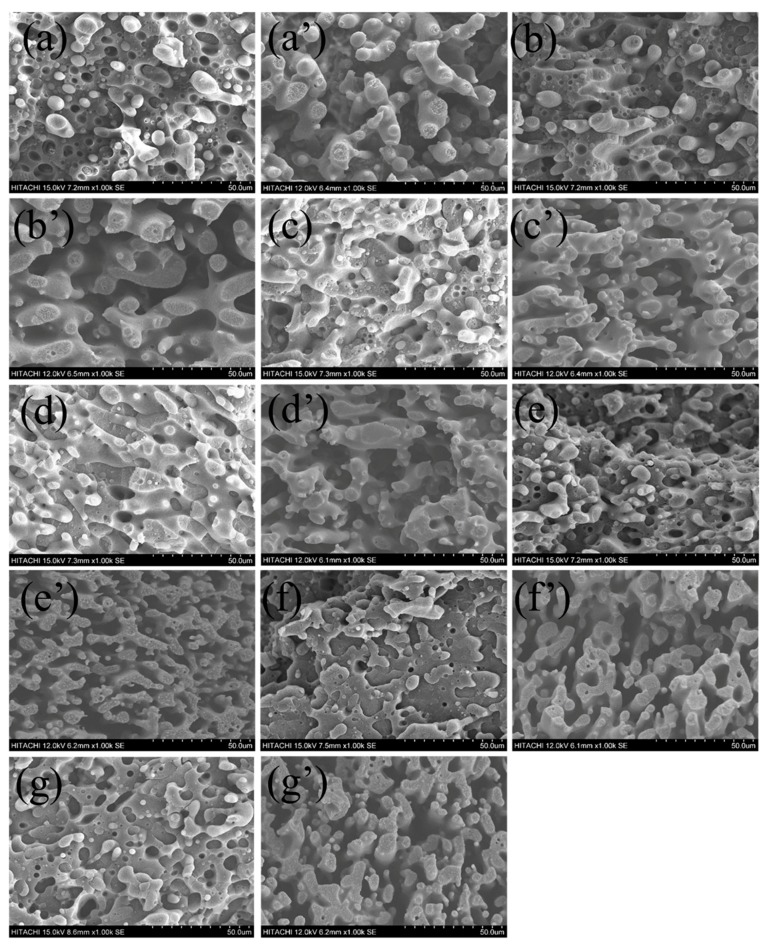
SEM of PLA/HDPE (70/30) blends with different carbon black (CB) loadings, (**a**,**a′**) 1.5 wt %, (**b**,**b′**) 3.0 wt %, (**c**,**c′**) 5.0 wt %, (**d**,**d′**) 7.5 wt %, (**e**,**e′**) 10.0 wt %, (**f**,**f′**) 12.5 wt %, and (**g**,**g′**) 15.0 wt %.

**Figure 3 polymers-11-01583-f003:**
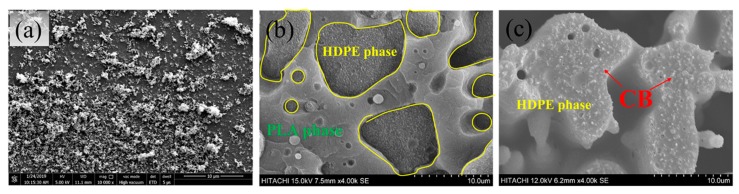
(**a**) SEM of CB, (**b**) SEM of PLA/HDPE/CB (70/30/10) blends, (**c**) SEM of PLA/HDPE/CB (70/30/10) blends etched PLA phase.

**Figure 4 polymers-11-01583-f004:**
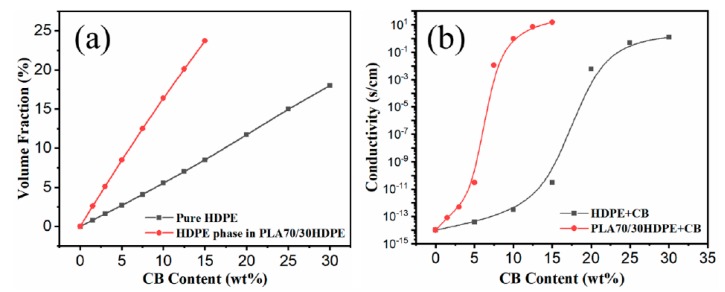
(**a**) The volume fraction of CB in the HDPE phase of PLA70/30HDPE blends and HDPE, (**b**) electrical conductivity of PLA70/30HDPE/CB blends and HDPE/CB blends.

**Figure 5 polymers-11-01583-f005:**
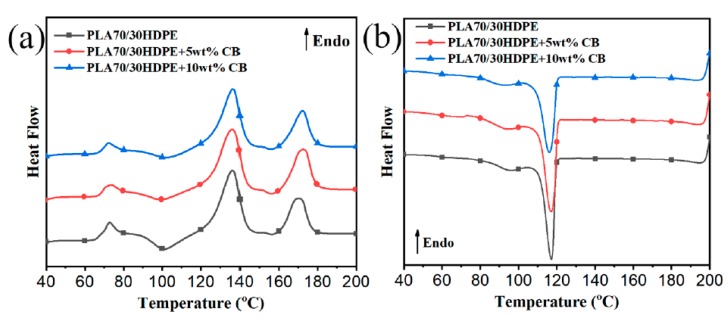
(**a**) Melting and (**b**) cooling curves of PLA70/30HDPE/CB with different CB loadings.

**Figure 6 polymers-11-01583-f006:**
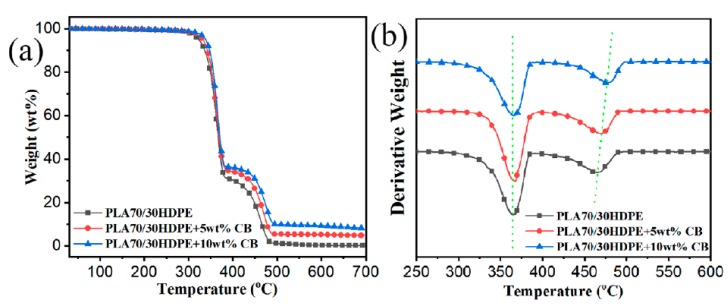
(**a**) TGA and (**b**) DTG curves of PLA70/30HDPE/CB with different CB loadings.

**Figure 7 polymers-11-01583-f007:**
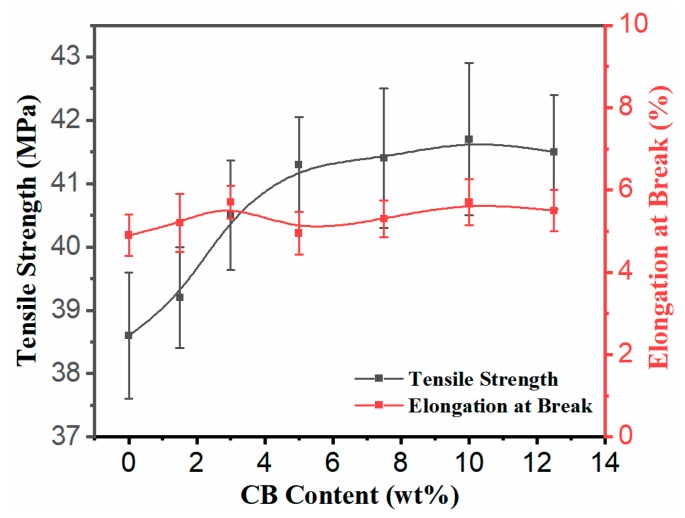
Tensile properties of PLA70/30HDPE/CB blends with different CB loadings.

**Figure 8 polymers-11-01583-f008:**
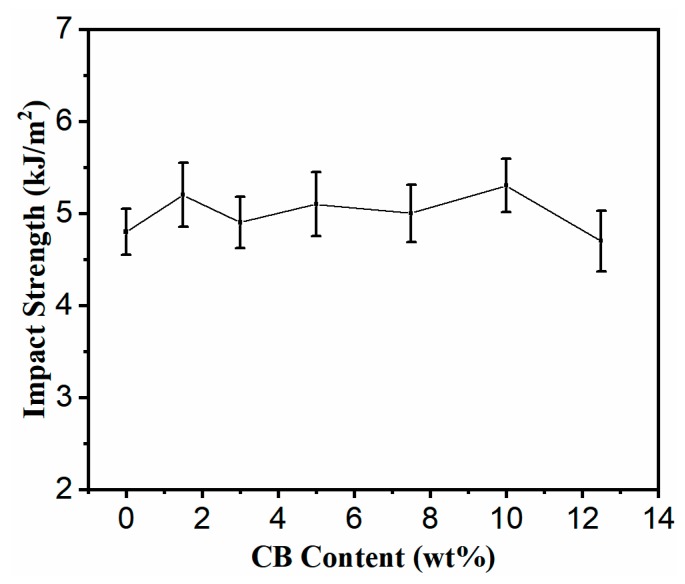
Impact strength of PLA70/30HDPE/CB blends with different CB loadings.

**Table 1 polymers-11-01583-t001:** Measured H_2_O and CH_2_I_2_ contact angles (at 25 °C) and calculated values of surface energy of the used high-density polyethylene (HDPE) and poly (lactic acid) (PLA). CB, carbon black.

Samples	$\theta_{H_{2}O}$ (°)	$\theta_{CH_{2}I_{2}}$ (°)	$\gamma^{d}$ (MJ/m^2^)	$\gamma^{p}$ (MJ/m^2^)	γ (MJ/m^2^)
CB	89.1 ± 1.0	45.8 ± 1.1	35.3	1.6	36.9
HDPE	87.9 ± 1.2	56.2 ± 1.0	28.5	2.1	30.6
PLA	84.2 ± 0.9	68.5 ± 0.8	16.5	8.7	25.2

**Table 2 polymers-11-01583-t002:** The thermal parameters of PLA70/30HDPE/CB with different CB loadings from differential scanning calorimeter (DSC).

CB Content	$T_{g-PLA}$ (°C)	$T_{m-PLA}$ (°C)	$T_{m-HDPE}$ (°C)	$T_{c-HDPE}$ (°C)	$∆H_{cc-PLA}$ (J/g)	$∆H_{m-PLA}$ (J/g)	$∆H_{m-HDPE}$ (J/g)	$X_{c-PLA}$ (%)	$X_{c-HDPE}$ (%)
0	70.6	168.3	136.6	117.6	10.2	29.1	59.4	29.0	67.8
5wt %	70.7	169.1	135.9	117.5	5.8	27.3	58.7	34.8	70.5
10wt %	70.5	168.7	136.5	116.8	2.3	26.2	57.5	40.7	72.9

**Table 3 polymers-11-01583-t003:** The thermal parameters of PLA70/30HDPE/CB with different CB loadings from thermogravimetric analysis (TGA).

CB Content	$T_{5}$ (°C)	$T_{max\left(PLA\right)}$ (°C)	$T_{max\left(HDPE\right)}$ (°C)	Charred Residues at 600 °C (wt %)
0	322.7	362.2	465.3	0.3
5wt %	332.2	363.5	471.7	5.2
10wt %	339.1	362.9	478.1	9.9

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
