# Peer review of "Selective Localization of Carbon Black in Bio-Based Poly (Lactic Acid)/Recycled High-Density Polyethylene Co-Continuous Blends to Design Electrical Conductive Composites with a Low Percolation Threshold"

_polymers, 2019, doi:10.3390/polym11101583_

Round 1

Reviewer 1 Report

Dear Authors,

thanks for the good paper

I had three remarks:

1) The paper has a lot mistakes related to the format:

Mostly subscript and superscript letters a not correct

In The paragraph with thermogravimetric analyse some brackets are also strange

The letters of the x-axis of figure 8 are big and therefore are not at right place.

Please control the format in th last version

2) microscopic pictures

Fig. 1 has different scales. The not-etched samples has 50 µm, the etched samples has 100 µm. In Fig. 2 it is inverse not-etched 100 µm, the etched samples 50 µm. Therefore, the pictures can not compared in good manner.

Please optimize this. Perhaps with same scales or with a table of the dimensions of the different polymers in the blend

3) Dispersion of CB in blend

In fig 3d I do not see the CB at the marked locations. Please explain it in a better way by adding some sentences because fig 3d looks similar to fig 1c. Otherwise it is not possible to demonstrate the dispersion of CB in PE. This fact is very important due to the explanation of the conductive measurement.

BR

The reviewer

Author Response

Thank you so much for your reviewing and comments on our manuscript. We do appreciate your reading and your approval on this work encourages us to make further improvement.

Reviewer 2 Report

The article aims a very interesting issue regarding the selective distribution of conductive particles in one of the components of a polymer blend in order to obtain particular properties (low percolation threshold). In principle, this topic is quite interesting as to be treated in one of the issues of the journal Polymers. In particular, the polymer system is based on the mixture of PLA with HDPE, with a particular composition that lead to co-continuous morphology.

Comments: 

1. Compatibility PLA/HDPE improved by adding CB. Explain ...

2. A deeper Analysis of results of conductivity is recommended. For example, is there any coincidence, as expected, between the electrical behavior of neat HDPE and PLA/HDPE when exclusively is taking into account the volume fraction of CB in the HDPE phase? Please check this and represent volume fraction of CB in HDPE but in the PLA/HDPE/CB as a function of conductivity to see if there is overlapping with the curve corresponding to the system PLA/CB. Discuss... Anything new was found??

3. Introduction, Lines 40-42 References are missed.

4. It would be necessary to describe how it is measured the contact angles, at least in the case of CB. Why there are not errors in the data of Table 1?

5. It would be interesting to have TEM or STEM images in order to better see selective location of CB. 

6. When talking about recycle HDPE its molecular weight is expected to be uncontrolled. However, when mixing incompatible polymers one of the main factors affecting the morphology is the molecular weight. In fact, blend with the same composition may lead to highly different morphologies depending on the molecular weights of the components. I is necessary to take into consideration this issue in the introduction and justify why in this particular work molecular weights were no taken into account.

7. The sentence: Lines 153-154 "The phase continuity of the co-continuous structure of PLA/HDPE/CB (70/30/x) ternary blend becomes better" should be explain. What do the authors want to say? what is exactly the improvement? It is stated that addition of CB increases the polymers compatibility; however, samples with different compositions in terms of CB lead to similar DSC traces with nearly same ratio between the endothermic peaks. There is not peaks overlapping when the compatibility increases nor peaks shift. Those experimental facts are usually indicative of compatibility improvement.

8. Attending the rule of mixtures, don't you expect other results for the mechanical properties??

Author Response

Thank you so much for your reviewing and comments on our manuscript. We do appreciate your reading and your approval on this work encourages us to make further improvement.

Q1: Compatibility PLA/HDPE improved by adding CB. Explain ....

A1: Thanks for your comment, Sorry, in this manuscript, the ability of CB to improve the compatibility of PLA/HDPE blends means that the introduction of CB reduces the phase size of the blend rather than increasing the interfacial interaction between PLA phase and HDPE phase. It is generally agreed that among different factors, the viscosity ratio (λ) (the ratio of the viscosity of the dispersed polymer to the viscosity of the matrix polymer) turns out to be one of the most critical variables in controlling the blend morphology. With the introduction of CB and the selective localization of CB in the HDPE component, the λ between HDPE and PLA was increased, and the blend morphology was changed. From the SEM images in our manuscript, the phase size of PLA/HDPE/CB composites decrease with the CB loading.

Q2: A deeper Analysis of results of conductivity is recommended. For example, is there any coincidence, as expected, between the electrical behavior of neat HDPE and PLA/HDPE when exclusively is taking into account the volume fraction of CB in the HDPE phase? Please check this and represent volume fraction of CB in HDPE but in the PLA/HDPE/CB as a function of conductivity to see if there is overlapping with the curve corresponding to the system PLA/CB. Discuss... Anything new was found??

A2: Thank you for your useful comment. We have added the detailed analysis of results of conductivity in the new manuscript according to your comment. As shown in Fig. 4 (a), the volume fraction of CB in HDPE/CB (75/25) composites is about 15 vol %, and the volume fraction of CB in the HDPE phase of PLA/HDPE/CB (70/30/10) composites is also about 15 vol %. At the same time, as shown in Fig. 4 (b), the electrical conductivity of HDPE/CB (75/25) and PLA/HDPE/CB (70/30/10) composites are all 10-1 s/cm level. The test results of HDPE/CB and PLA/HDPE/CB (70/30/x) composites with the same CB volume fraction in HDPE component are in good agreement. It indicates that the co-continuous micro structure and CB selective localization is a good approach to prepare the electrical conductive polymer composites with low electrically conductive percolation threshold.

Q3: Introduction, Lines 40-42 References are missed.

A3: According to your suggestion, we have added the References in Lines 40-42.

Q4: It would be necessary to describe how it is measured the contact angles, at least in the case of CB. Why there are not errors in the data of Table 1?

A4: The question raised by the reviewer is important. Contact angle tests were performed with an OCA40 apparatus (Dataphysics Co., Ltd., Germany), static contact angles of distilled water were measured by depositing a drop of 3-5 mL on the sample surface, and the values were estimated as the tangent normal to the drop at the intersection between the sessile drop and the surface. In table 1, all the date is the average value for three repeats, so the errors is not listed. In addition, PLA and HDPE samples for contact angle measurement were compression-molded between clean polyester films at 190 oC for 4min and then cooled to 25 oC under pressure for 1 min. CB sample for contact angle measurement was compression-molded between PTFE films at ambient temperature under pressure for 3 min.

Q5: It would be interesting to have TEM or STEM images in order to better see selective location of CB.

A5: Thank you for your useful comment. The TEM and STEM is the better way to see the selective location of CB. But in this manuscript, the SEM and contact angle tests have fully demonstrated the selective dispersion of CB in HDPE phase. In addition, according to the repair time (10 days) given by the editor, we can not provide the TEM and STEM images in our new manuscript.

Q6: When talking about recycle HDPE its molecular weight is expected to be uncontrolled. However, when mixing incompatible polymers one of the main factors affecting the morphology is the molecular weight. In fact, blend with the same composition may lead to highly different morphologies depending on the molecular weights of the components. I is necessary to take into consideration this issue in the introduction and justify why in this particular work molecular weights were no taken into account.

A6: Thank you for your useful comment. Actually, for polymer blends, the viscosity ratio (λ) (the ratio of the viscosity of the dispersed polymer to the viscosity of the matrix polymer) is one of the most critical variables in controlling the blend morphology. And as you said, the molecular weight and distribution of HDPE will directly determine the viscosity of HDPE. Nevertheless, the testing process of molecular weight and its distribution is cumbersome. For convenience, the viscosity, especially the melt flow index (MFI), is often used to indirectly characterize the molecular weight and distribution of polymer in the industry. Thus, we just obtain the MFI value of the recycled HDPE from the provider.

Q7: The sentence: Lines 153-154 "The phase continuity of the co-continuous structure of PLA/HDPE/CB (70/30/x) ternary blend becomes better" should be explain. What do the authors want to say? what is exactly the improvement? It is stated that addition of CB increases the polymers compatibility; however, samples with different compositions in terms of CB lead to similar DSC traces with nearly same ratio between the endothermic peaks. There is not peaks overlapping when the compatibility increases nor peaks shift. Those experimental facts are usually indicative of compatibility improvement.

A7: Thank you for this good comment. As describe in A1, the ability of CB to improve the compatibility of PLA/HDPE blends means that the introduction of CB reduces the phase size of the blend rather than increasing the interfacial interaction between PLA phase and HDPE phase. However, with the introduction of CB, although the phase size of the PLA/HDPE/CB blendS is reduced, PLA component and HDPE component are still incompatible in the molecular scale due to the differences in molecular structure. Thus, there is not peaks overlapping in the DSC and TGA curves.

Q8: Attending the rule of mixtures, don't you expect other results for the mechanical properties??

A8: Thank you for your useful comment. Attending the rule of mixtures, there should be more interesting performance discoveries for polymer blends with co-continuous structure. However, due to the space limitations, we have not been able to discuss in detail the other interesting features in this manuscript.

Round 2

Reviewer 1 Report

Thanks for your revision

For my point of view everything is explained in a good manner

BR

The reviewer

Author Response

Thank You!

Reviewer 2 Report

Changes over the first version should be highlighted Even with three measurements errors should be given. AS DSC result revealed, the use of the term compatibility is not correct in this manuscript. Changes observed should be described in terms of morphology.

Author Response

Q1: Changes over the first version should be highlighted.

A1: First, we thank the reviewer for his/her very positive comments to our manuscript. Following the reviewer suggestion, all the changed part have been highlighted in the second version.

Q2: Even with three measurements errors should be given.

A2: Thank you for this good comment. And all the measurements errors were given and highlighted in the revised version.

Q3: AS DSC result revealed, the use of the term compatibility is not correct in this manuscript. Changes observed should be described in terms of morphology.

A3: Thank you for your useful comment. We have replaced the use of the term compatibility in the new revised manuscript.